# Development of Carbon Nanotube/Silicone Pad for Improved Performance of Electromyostimulation Training

**Myeongcheol Kim [1], Jaegyun Im [2], Junsul Park [1], Kyung-Bum Kim [3],\* and Jaegeun Lee [2],\***

1   Coremovement, BI Center 404, Sinseon-ro 365, Nam-gu, Busan 46241, Korea
2   School of Chemical Engineering, Pusan National University, 2 Busandaehak-ro 63 beon-gil, Geumjeong-gu, Busan 46241, Korea
3   Research Institute of Industrial Technology, Pusan National University, 2 Busandaehak-ro 63 beon-gil, Geumjeong-gu, Busan 46241, Korea
*   Correspondence: lnylove17@pusan.ac.kr (K.-B.K.); jglee@pusan.ac.kr (J.L.)

**Abstract:** We have developed a carbon nanotube (CNT) pad to replace carbon black, which is essential for electric muscle stimulation (EMS) suits that can provide efficient exercise effects in a short time. The optimized CNT pad had 10 times lower concentration but showed 20 times lower resistance than the carbon black pad. In the case of the peak voltage indicating the EMS performance, it was confirmed that the CNT (4.0 wt%) was 25.9 V and the carbon black (40 wt%) was 6.5 V, which was about 4 times better. CNT added increased from 4.0 wt% to 10.0 wt%, and the peak voltage increased from 25.9 V to 26.8 V, but the output voltage was not significantly improved compared to the amount of CNT added. These experimental results are expected to show higher EMS properties than carbon black because carbon nanotubes and silicon are agglomerated to form a particle-like shape.

**Keywords:** electrical muscle stimulation; carbon nanotube; carbon black

## 1. Introduction

Electrical muscle stimulation (EMS) is known to induce contraction of muscles by applying an electric current to the muscle through electrodes. The human sensory system sends electrical impulses to the cranial nerves periodically to apply an electric current to the nerve roots to trigger contractile activity [1,2]. EMS supports one to reach high-intensity exercise levels that are difficult to achieve with normal exercise. EMS was mainly used for muscle weakness or treatment and recovery. The global home fitness equipment market size is projected to grow at USD 657.4 million during 2020–2024, growing at a CAGR of over 3% during the forecast period. The development of wearable device technology and the growth of the health-related fitness industry are spreading as a 'person-centered smart health' trend. Changes in the external exercise environment such as fine dust and the prevalence of viral infectious diseases such as Corona 19 are expected to advance the transition to the home training-based fitness industry [3–5]. Recently, the implementation of EMS in systematically screened clinical trials has resulted in significant improvements in muscle strength, exercise capacity, and disease-specific health status [6,7]. Diseases associated with insulin resistance, such as type 2 diabetes, and certain types of cancer, particularly hormone-related cancers, colorectal cancer, and gallbladder disease are associated with a high body fat percentage status and decreased muscle mass [8].

Unlike general fitness equipment, it is connected to the equipment by wearing an exercise suit using low-frequency microcurrent to maximize muscle strength. Furthermore, EMS provides an opportunity for haptic feedback in virtual space. Virtual reality (VR) games enhance storytelling by delivering output in the form of visual, auditory, or tactile feedback [9,10]. Due to the lack of interaction in the form of user input, improved feedback can benefit significantly. While the user maintains a passive state, the VR game is utilized to induce the user's physical movement with various body parts to correspond to the

movement of the individual avatar [11]. This kind of feedback can explore the potential of EMS as a haptic feedback technology in wearable devices [12,13].

EMS technology does not require moving mechanical parts and can be easily miniaturized because it is driven by the electric current generated by the carbon/silicone pad. So far, carbon blacks have been used as conducting additives in the pad. Carbon blacks are electrically conductive, mechanically strong, and cheap, so they have been widely used as conductive additives in many applications. However, the spherical morphology of carbon blacks is not beneficial for forming a conducting network. Carbon nanotubes (CNTs) are ideal alternatives for carbon blacks since CNTs have a one-dimensional structure and are thus more advantageous for forming a conducting network with a much small amount of addition. In this paper, we report on the development of the CNT/silicone pad for the improvement of EMS performance. We demonstrate that the EMS performance of the CNT/silicone pad measured by resistance and output voltage is superior to that of the conventional carbon black/silicone pad. The study shows that EMS technology has great potential to be improved by the development of the hardware.

## 2. Experimental Section

We fabricated CNT pads were fabricated using multi-walled CNTs (LG, BT1001M, Seoul, Korea). For comparison of EMS characteristics, pads were manufactured using carbon black (OCI, N550S, Seoul, Korea). Commercially available liquid polydimethylsiloxane (PDMS, Dow Corning, Sygard 184, Midland, TX, USA) is used to mix 1.0 to 10.0 wt% of CNT powder. A CNT pad was prepared by curing PDMS + CNT in a liquid state for 1 h in a mold for manufacturing an EMS suit.

The conductive fiber of the suit is made in the form of a pad of silicon material mixed with CNT. The pad microcurrent control unit is characterized in that it comprises a circuit for controlling the oscillation frequency of the current provided from the power source and a circuit for controlling the duration of the frequency.

By controlling the frequency of the microcurrent, the form of stimulation felt by the user can be varied in various ways, so it has the effect of realizing various stimuli according to situations such as virtual reality to obtain a perfect sense of reality.

The EMS characteristics are measured through the electrical output test of the pad to which different amounts of CNT are added, and the output voltage generated at the pad is checked through an oscilloscope. Pads with different CNT addition amounts generate different output voltages according to the added amounts. The output voltage of the pad was measured using a voltage generator (Yokogawa 2558A AC Voltage Current Standard, Tokyo, Japan), 100 $V_{PP}$ was applied to the pad, and the voltage coming out of the pad was measured using an oscilloscope (SIGLENT SDS 1102CLM+, Helmond, Netherlands).

## 3. Result and Discussion

Figure 1a shows actual images of a pad with CNT 1.0, 2.5, 4.0, 5.0, and 10.0 wt% added and a carbon black pad. The electrical resistance of each pad was measured and marked. The resistance of the CNT 1.0 and CNT 2.5 pads was not measured, the resistance of the CNT 4.0 pad was measured to be 0.04 kΩ and that of the CNT 5.0 to 0.03 kΩ. The resistance of the CNT 10.0 pad with the highest CNT content was measured to be 0.028 kΩ. In Figure 1b, the microstructure of the pad was observed by adding carbon black and CNT, and the correlation with the electrical resistance was revealed through the difference. In the case of the carbon black pad, angular carbon was clearly observed and silicon was partially confirmed. In the case of CNT 1.0 and 2.5, aggregation was not observed on the pad surface, and a flat appearance was exhibited. Aggregation started gradually from CNT 4.0, and aggregation in the form of clear grains was observed from CNT 5.0 and CNT 10.0 pads.

Figure 2a observes the carbon component of the pad due to the addition of CNTs and tries to reveal the correlation with the electrical resistance through the difference. In the case of the carbon black pad, a carbon component of 61.2 wt% was observed, and as the CNT content increased from 1.0 to 2.5, the carbon content was 16.4. In terms of weight ratio,

it showed a tendency to increase by 19.6 wt%, and it was found that as the CNT content increased from 4.0 to 10.0, it increased from 27.3 wt% to 36.4wt%. Compared to carbon black, the carbon content of the CNT 10.0 pad decreased from 61.2 wt% to 36.4 wt%, but the electrical resistance decreased 7 times from 0.2 kΩ to 0.028 kΩ. It was confirmed that the carbon content of the CNT 10.0 pad had lower electrical resistance than the high carbon content of carbon black. Figure 2b observes the silicon component of the pad according to the addition of carbon black and CNT, respectively, and reveals the correlation with the electrical resistance through the difference. In the case of the carbon black pad, a silicon component of 38.8 wt% was observed, and in the case of CNT 1.0 and 2.5, the silicon content decreased from 56.2 wt% to 54.9 wt% as the addition amount increased. As the CNT increased from 4.0 to 10.0, a phenomenon was found that decreased from 50.2 wt% to 37.3 wt%. Compared with carbon black (Si k: 38.8 wt%), the silicon content of the CNT 10.0 pad (Si k: 37.3 wt%) is 1.5 wt%, which is not much different, but the electrical resistance is about 7 times lower.

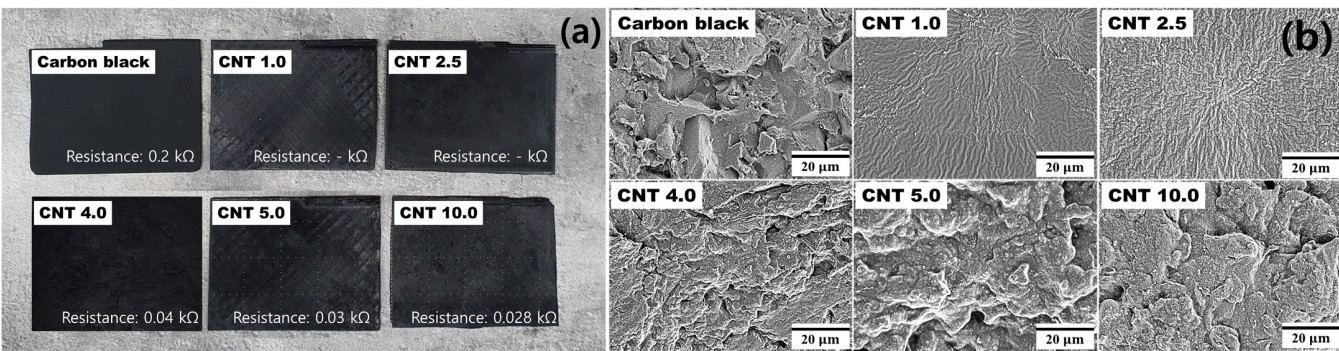

**Figure 1.** (**a**) EMS pads according to the amount of commercialized carbon black and preparation CNT added, (**b**) Microstructure images using FE-SEM of the pad by carbon black and CNT addition.

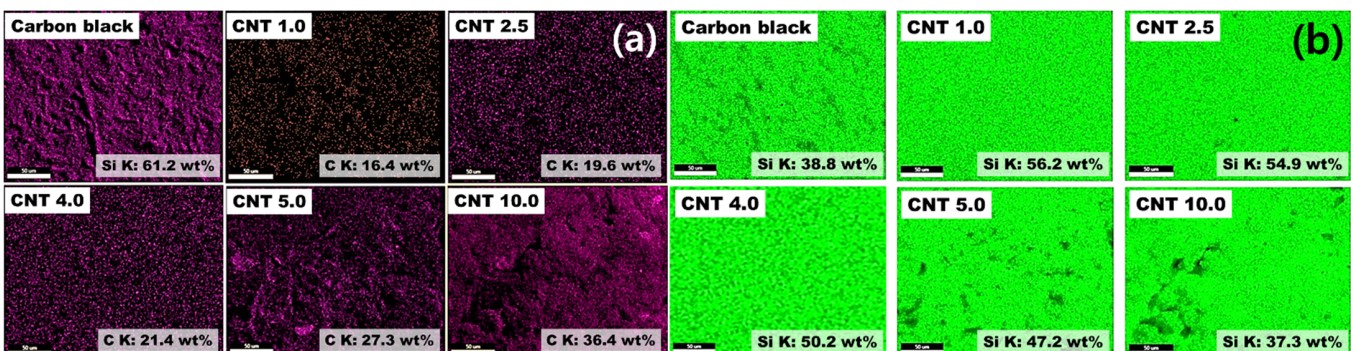

**Figure 2.** (**a**) Carbon component mapping image using EDS of the pad for each carbon black and CNT addition, (**b**) Silicon component mapping image using EDS of the pad by carbon black and CNT addition.

Figure 3 shows the correlation between the carbon component of the pad, the weight of silicon, and the electrical resistance according to the addition of carbon black and CNT. As the amount of CNT added increases, silicon tends to decrease and carbon tends to increase. As a result, the electrical resistance is reduced, and CNT 4.0 (electrical resistance: 0.04 kΩ) has a carbon content of 21.4 wt% compared to carbon black (electrical resistance: 0.2 kΩ) with a carbon content of 60 wt% higher, which is relatively 5 compared to the carbon of carbon black. It has twice the carbon content. Detailed data are shown in Table 1 below.

Figure 4 compares the electrical output of the pad for each carbon black and CNT addition. For carbon black, the peak voltage was measured to be 6.5 V. For the CNT 1.0 pad, a 0.0 V peak voltage was measured, and for the CNT 2.5 pad, the peak voltage slightly increased to 4.6 V. In the case of the CNT 4.0 pad, the peak voltage characteristic rapidly

increased to 25.9 V. Compared to carbon black, the peak voltage was measured to be about 3.98 times higher than that of carbon black. Similar peak voltages of 26.7 V and 26.8 V were observed for CNT 5.0 and CNT 10.0 pads. In the case of CNT 4.0 with a sharply increased peak voltage, carbon and silicon began to aggregate more than 30 μm, and the output voltage was measured to be 3.98 times higher than that of carbon black. However, as the amount of CNT added increased from 4.0 wt% to 10.0 wt%, the peak voltage increased from 25.9 V to 26.8 V, but the output voltage was not significantly improved compared to the amount of CNT added.

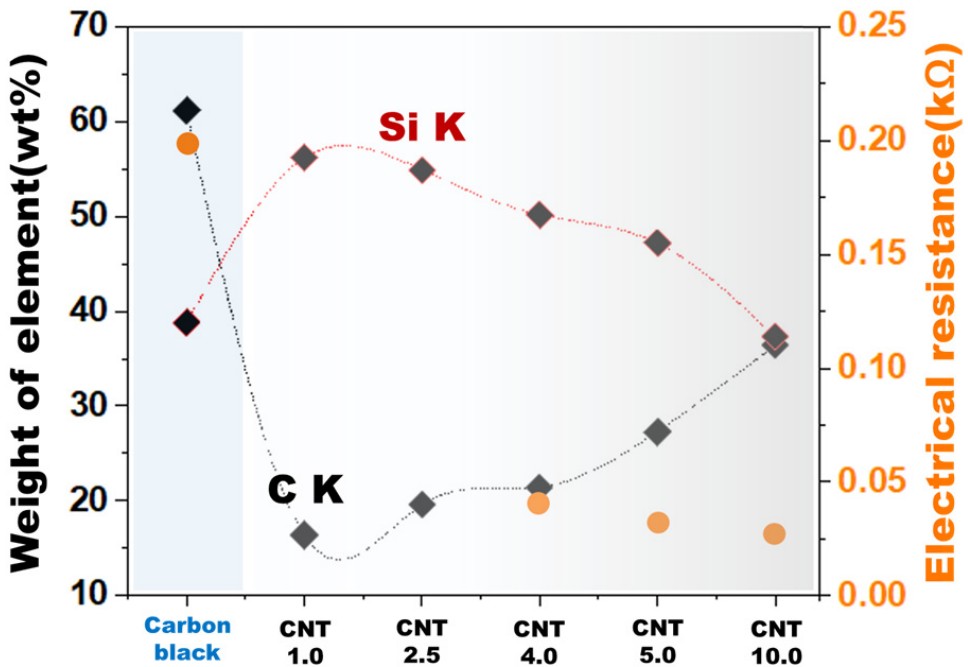

**Figure 3.** Correlation between the weight and electrical resistance of silicon and carbon components of the pad by carbon black and CNT addition.

**Table 1.** Electrical resistance, silicon and carbon component weight ratio according to carbon black and CNT addition amount.

|  | Component (wt%) | Resistance (kΩ) | Weight of Si (wt%) | Weight of C (wt%) |
|---|---|---|---|---|
| Carbon black | 40 | 0.2 | 38.8 | 61.2 |
| CNT 1.0 | 1.0 | - | 56.2 | 16.4 |
| CNT 2.5 | 2.5 | - | 54.9 | 19.6 |
| CNT 4.0 | 4.0 | 0.04 | 50.2 | 21.4 |
| CNT 5.0 | 5.0 | 0.03 | 47.2 | 27.3 |
| CNT 10.0 | 10.0 | 0.028 | 37.3 | 36.4 |

Figure 5 graphically shows the electrical resistance caused by the change of the pad's microsurface according to the amount of carbon black and CNT added and the change in the output voltage that makes the skin feel a vibration. Through the graph of each CNT pad between the electrical resistance and the output voltage, it was confirmed that the output voltage rapidly increased when the electrical resistance was measured to be 0.04 kΩ or more. In the case of CNT 4.0, where the peak voltage rapidly increased, carbon and silicon began to agglomerate over 30 μm, CNT 1.0 and 2.5 had a flat microstructure, and the output voltage of CNT 4.0 was 3.98 times that of carbon black. From the time when silicon and CNT aggregation started, the resistance decreased rapidly and the peak voltage of EMS also increased rapidly. However, in the case of CNT 5.0 and CNT 10.0 added over CNT 4.0, it was confirmed that there was no significant difference between CNT 4.0 and

CNT 10.0 with a 0.9 V difference with an output voltage of 25.9–26.8 V. Therefore, in this study, when CNT 4.0 pad was applied to EMS, it was observed that the electrical efficiency was higher than that of carbon black, the measured data are shown in Table 2.

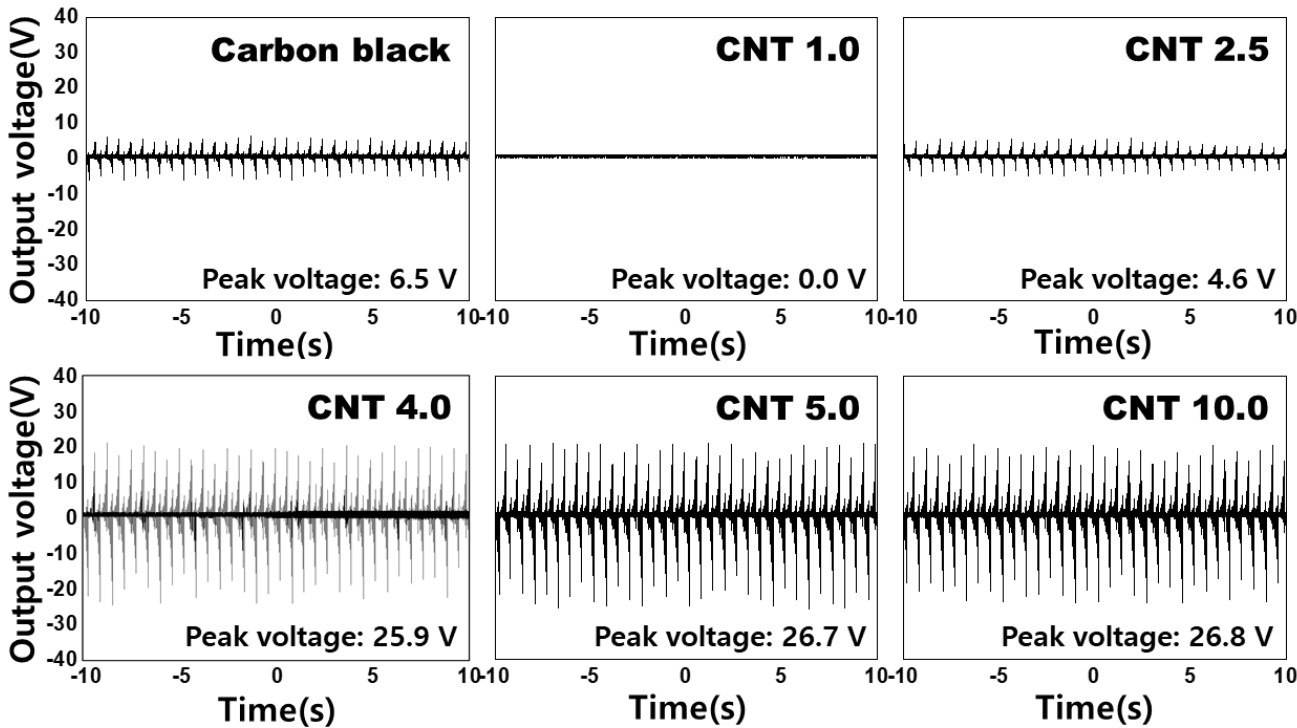

**Figure 4.** Electrical output graph of the pad by carbon black and CNT addition.

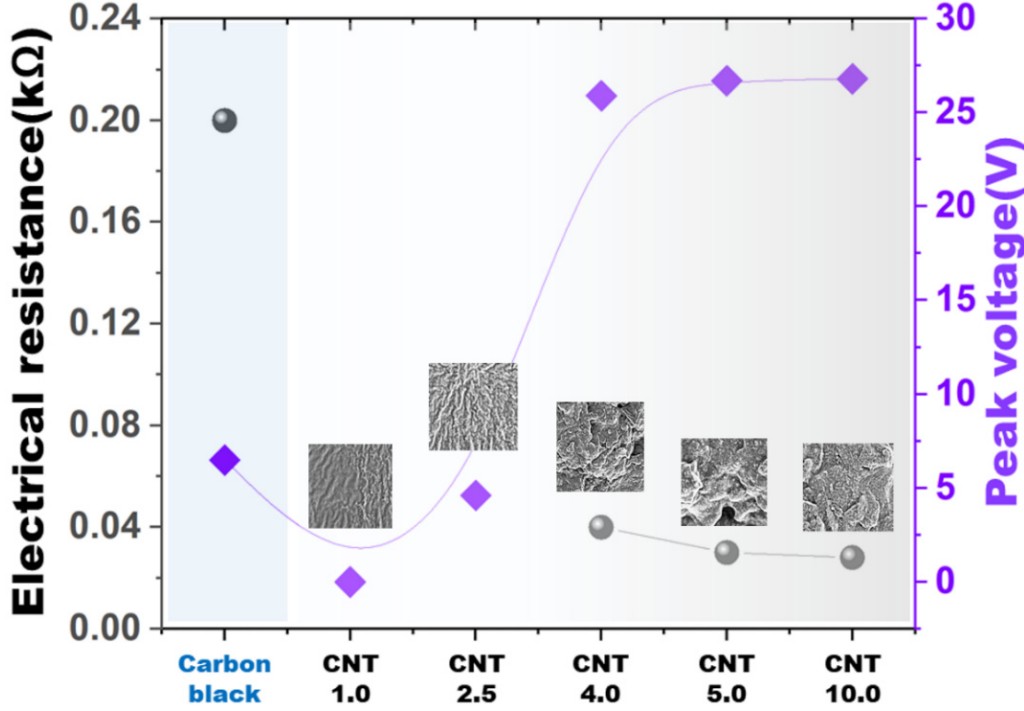

**Figure 5.** Correlation graph between the electrical resistance and output voltage of the pad for each carbon black and CNT addition.

**Table 2.** Data on the electrical resistance and output voltage of the pad for each carbon black and CNT addition.

|  | Component (wt%) | Peak Voltage (V) | Resistance (kΩ) |
|---|---|---|---|
| Carbon black | 40 | 6.5 | 0.2 |
| CNT 1.0 | 1.0 | 0 | - |
| CNT 2.5 | 2.5 | 5.8 | - |
| CNT 4.0 | 4.0 | 25.9 | 0.04 |
| CNT 5.0 | 5.0 | 26.7 | 0.03 |
| CNT 10.0 | 10.0 | 26.8 | 0.028 |

## 4. Conclusions

The CNT pad confirmed the possibility of replacing the carbon black pad, which is essential for the electric muscle stimulation (EMS) suit. When the amount of CNT added was 4.0 wt%, the resistance was measured to be 20 times or less compared to carbon black (40 wt%). In the case of the peak voltage indicating the EMS characteristic value, carbon black showed an output value of 6.5 V and a CNT output value of 25.9 V, confirming that the performance was about 4 times superior. These experimental results were observed to exhibit improved EMS properties due to the agglomeration phenomenon that has high conductivity and high density of CNTs.

**Author Contributions:** Data curation, J.P.; Investigation, K.-B.K.; Resources, M.K., J.I. and J.L. All authors have read and agreed to the published version of the manuscript.

**Funding:** This research was supported by the technology transfer and commercialization Program through INNOPOLIS Foundation funded by the Ministry of Science and ICT,(2021-BS-RD-0131/AI-based exercise coaching and feedback system using new haptic suit technology), This work was supported by Pusan National University Research Grant, 2019.

**Conflicts of Interest:** The authors declare no conflict of interest.

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
