# Peer review of "Development of Carbon Nanotube/Silicone Pad for Improved Performance of Electromyostimulation Training"

_energies, doi:10.3390/en15186681_

Round 1

Reviewer 1 Report

1. Please explain the meaning of “Electromyostimulation Traing” in detail.

2. Some grammar mistakes should be revised. For example, in line 32, “High body 32 fat percentage conditions associated with insulin resistance, such as type 2 diabetes, cer-33 tain types of cancer, particularly hormone-related cancers, colorectal cancer and gallblad-34 der disease, a decrease in muscle mass is associated with a decrease in muscle strength.”

3. Please demonstrate the novelty of this work in the introduction.

4. The application of the proposed material should be investigated as well.

5. Please pay attention to the problems in layout. For example, Figure 1 and its legend should be in the same page.

Author Response

1. Please explain the meaning of “Electromyostimulation Traing” in detail.

Training was corrected to training due to a typo.

“Electrical stimulation training is a widely used methodology in applied sports science. Unlike normal voluntary contracts initiated by the central nervous system (eg resistance training), EMS involves involuntary contracts induced by electrical currents applied to muscles.”

2. Some grammar mistakes should be revised. For example, in line 32, “High body 32 fat percentage conditions associated with insulin resistance, such as type 2 diabetes, cer-33 tain types of cancer, particularly hormone-related cancers, colorectal cancer and gallblad-34 der disease, a decrease in muscle mass is associated with a decrease in muscle strength.”

Edited to “Diseases associated with insulin resistance, such as type 2 diabetes, certain types of cancer, particularly hormone-related cancers, colorectal cancer, and gallbladder disease are associated with a high body fat percentage status and decreased muscle mass.”

3. Please demonstrate the novelty of this work in the introduction.

In the case of the peak voltage, an indication of the performance of EMS, the CNT/silicone pad showed 25.9 V while the carbon black/silicone pad showed 6.5 V, confirming that the performance was approximately 4 times superior.

4. The application of the proposed material should be investigated as well.

There is no pad technology using CNT related to EMS technology, and most of them are using carbon black.

5. Please pay attention to the problems in layout. For example, Figure 1 and its legend should be in the same page.

The text has been modified to fit the format.

Reviewer 2 Report

The authors propose replacing traditional carbon black/silicone with carbon nanotubes/silicone (CNT/silicone). It has been shown that CNT/silicone makes it possible to significantly improve materials with a smaller addition of silicone compared to commercially available materials, which should affect (reduce) the final cost of the product. Insufficient description of the problem and the need to solve it do not allow us to highly appreciate the significance of the ongoing research. Incorrect presentation and the presence of errors confuse the reviewer.

1. The authors need to remove all typos from the text of the article, which are present, for example, on page 2 in line 53, two dots; page 3 lines 104–105 there are no spaces between the digit and the unit of measure, and so on through the text.

2. It is necessary to unify the list of sources used, and according to the rules of the journal.

3. Most of the used literature is 2005-2008, it is necessary to update the list of references with newer references.

4. If the use of carbon nanotubes is justified by their one-dimensionality, then why not use graphene sheets? A clearer rationale for the choice of CNT is needed.

5. A comparative study of the durability of the carbon/silicone materials used would be useful, perhaps the stability of samples using CNT would be higher than Carbon black.

6. Problems with the English language, in some sentences there are no predicates:

- in Abstract the sentence "Comparison with carbon black pad on sale by adding 1.0-10.0 wt% of ..." (lines 13–14)

- in the Introduction (lines 32-35) the sentence "High body fat percentage conditions associated with insulin..." etc.)

- sentences that are not related in meaning ("...measured, and the resistance of the CNT 4.0 pad was 0.04 kΩ and CNT 5.0 was measured to be 0.03 kΩ." (lines 76–78)

- and others

All this significantly complicates the understanding of the article. The absence of articles and tense agreement (sentences on lines 104-107, 115-118) also does not make it possible to understand the meaning of the text.

8. The experimental part must be expanded with a detailed description of the use of methods and the applied formulas for calculating the electrical resistance indicator.

9. Information on Resistance is repeated in the text, table 1 and table 2, it must be left either in the text of the article or in one of the tables. Similar remark on Component (tables 1 and 2).

10. The conclusion summarizes the work done, but does not make it possible to understand how these results will affect the solution of the problem as a whole.

11. From the text of the article it is not entirely clear how Component and Weight of Si or Weight of C differ, a more detailed division is needed.

12. It is necessary to determine the surface of materials using the BET method. Structural studies like this will further point to the reasons for the lower Resistance values and higher Peak voltage values compared to commercial carbon black/silicone.

Author Response

1. The authors need to remove all typos from the text of the article, which are present, for example, on page 2 in line 53, two dots; page 3 lines 104–105 there are no spaces between the digit and the unit of measure, and so on through the text.

It has been modified according to the opinion of the judges.

2. It is necessary to unify the list of sources used, and according to the rules of the journal

It has been modified according to the opinion of the judges.

3. Most of the used literature is 2005-2008, it is necessary to update the list of references with newer references.

It has been modified according to the opinion of the judges.

4. If the use of carbon nanotubes is justified by their one-dimensionality, then why not use graphene sheets? A clearer rationale for the choice of CNT is needed.

In terms of conductivity, CNT is very advanced, and in the case of graphene, when mass-produced, conductivity is low due to oxide.

5. A comparative study of the durability of the carbon/silicone materials used would be useful, perhaps the stability of samples using CNT would be higher than Carbon black.

6. Problems with the English language, in some sentences there are no predicates:

It has been modified according to the opinion of the judges.

7. The experimental part must be expanded with a detailed description of the use of methods and the applied formulas for calculating the electrical resistance indicator.

This is the result of measuring the resistance of the same size for all samples 5 times using the 5-pin method and averaging the value.

8. Information on Resistance is repeated in the text, table 1 and table 2, it must be left either in the text of the article or in one of the tables. Similar remark on Component (tables 1 and 2).

Resistance values ​​in Tables 1 and 2 are both important factors and were used as data to be compared with each property.

9. The conclusion summarizes the work done, but does not make it possible to understand how these results will affect the solution of the problem as a whole.

This is the part that proves with data that it is an EMS pad material that can replace carbon black due to the high electrical conductivity of CNT.

10. From the text of the article it is not entirely clear how Component and Weight of Si or Weight of C differ, a more detailed division is needed.

The EDS data showed that the pad resistance decreased as the amount of C K added increased.

11. It is necessary to determine the surface of materials using the BET method. Structural studies like this will further point to the reasons for the lower Resistance values and higher Peak voltage values compared to commercial carbon black/silicone.

BET could not be measured due to the lack of research equipment, but it can be fully explained through the layering of C K content through EDS and the aggregation phenomenon through SEM.

Round 2

Reviewer 1 Report

The paper titled "Development of carbon nanotube/silicone pad for improved performance of electromyostimulation traing" has been mainly revised with necessary corrections and amendments according to the reviewers' comments, and the correlative parts have been rewrote by adding more information. The resulting manuscript is more satisfying. In my opinion, the paper could be published as is.

Reviewer 2 Report

The article can be accepted in its current form